# The crystal structure of bromide-bound *Gt*ACR1 reveals a pre-activated state in the transmembrane anion tunnel

Hai Li[1†], Chia-Ying Huang[2†], Elena G Govorunova[1], Oleg A Sineshchekov[1], Adrian Yi[3], Kenneth J Rothschild[3], Meitian Wang[2], Lei Zheng[1*], John L Spudich[1*]

[1]Department of Biochemistry and Molecular Biology, Center for Membrane Biology, University of Texas Health Science Center – McGovern Medical School, Houston, United States; [2]Swiss Light Source, Paul Scherrer Institute, Villigen, Switzerland; [3]Molecular Biophysics Laboratory, Photonics Center and Department of Physics, Boston University, Boston, United States

**Abstract** The crystal structure of the light-gated anion channel *Gt*ACR1 reported in our previous Research Article (Li et al., 2019) revealed a continuous tunnel traversing the protein from extracellular to intracellular pores. We proposed the tunnel as the conductance channel closed by three constrictions: C1 in the extracellular half, mid-membrane C2 containing the photoactive site, and C3 on the cytoplasmic side. Reported here, the crystal structure of bromide-bound *Gt*ACR1 reveals structural changes that relax the C1 and C3 constrictions, including a novel salt-bridge switch mechanism involving C1 and the photoactive site. These findings indicate that substrate binding induces a transition from an inactivated state to a pre-activated state in the dark that facilitates channel opening by reducing free energy in the tunnel constrictions. The results provide direct evidence that the tunnel is the closed form of the channel of *Gt*ACR1 and shed light on the light-gated channel activation mechanism.

**\*For correspondence:**
Lei.Zheng@uth.tmc.edu (LZ);
John.L.Spudich@uth.tmc.edu (JLS)

[†]These authors contributed equally to this work

## Introduction

*Gt*ACR1 is a light-gated anion channel discovered in 2015 (*Govorunova et al., 2015*) now widely used in optogenetics as a neuron-silencing tool. *Gt*ACR1 conducts both bromide and chloride ions effectively with higher relative permeability for the former substrate (*Govorunova et al., 2015*). We and the group of Karl Deisseroth independently determined X-ray crystal structures of the dark (closed) form of *Gt*ACR1 homodimer at 2.9 Å (*Kim et al., 2018*; *Li et al., 2019*). We proposed that the conductance pathway was attributable to a full-length intramolecular tunnel traversing each protomer from the extracellular side to the intracellular side of the membrane lined by mostly hydrophobic residues (*Li et al., 2019*). Current rectification by charges introduced inside but not outside the tunnel support our hypothesis that the tunnel serves as the anion-conducting path upon photoactivation (*Sineshchekov et al., 2019*). However, no substrate was found in either structure (*Kim et al., 2018*; *Li et al., 2019*) despite the presence of chloride in the crystallization conditions. The mechanism of anion conductance is still elusive.

In the apo form (i.e., without anion substrate) (*Li et al., 2019*) the tunnel is narrowed by three constrictions blocking ion permeation: Constriction 1 (C1) in the extracellular half, C2 mid-membrane consisting of the photoactive retinylidene Schiff base and interacting residues, and C3 in the cytoplasmic half. In our model, retinal photoisomerization at C2 needs to open all three of these gates to form a conductive anion channel through the protein. However, structural changes at the two constrictions, C1 and C3, which are located on each side of the Schiff base, and their roles in the channel gating mechanism are unclear.

To address these questions, here we report the crystal structure of bromide-bound *Gt*ACR1. The structure in bromide provides direct evidence for our proposed conductance mechanism (*Li et al., 2019*) and also demonstrates protein conformational changes in C1 and C3, which shed light into the role of the constrictions in channel opening.

## Results

### Overall structure of bromide-bound *Gt*ACR1

Bromide-bound *Gt*ACR1 crystals were obtained in the same conditions and pH used for the apo form structure, except for replacement of NaCl with NaBr in both protein purification and crystallization buffers. Avoiding radiation damage, small wedge data sets were collected using the *in meso in situ* serial data collection method (IMISX) (*Huang et al., 2018*). A detailed description of data collection and processing is provided in 'Materials and methods'. The structure of bromide-bound *Gt*ACR1 was determined at 3.2 Å resolution by molecular replacement (MR) using the apo form structure (PDB code 6EDQ) as the model (*Table 1*).

The bromide-bound *Gt*ACR1 exhibits a similar homodimeric overall structure as the apo form (rmsd: ~0.6 Å by comparing Cα from residues 1 to 295) (*Figure 1A*). All 7-helices and *trans*-configured retinal moieties are well superimposed, including the inter-subunit disulfide bridge stabilizing the N-terminus fragments of the two protomers on the extracellular surface. The two protomers, chain A and chain B, are also well superimposed with an rmsd of ~0.5 Å. Each protomer exhibits a continuous tunnel extending from the extracellular to intracellular surfaces of the protein similar to that seen in the apo form structure (*Figure 1A*).

As seen from comparison of the tunnel profile diameters of the apo- and bromide-bound structures (*Figure 1B–C*), bromide binding enlarges the tunnel in constriction C3 on the cytoplasmic side of the tunnel, where it is bound (*Figure 1A*), and also in part of C2, adjacent to the binding site, and the more distant constriction C1 on the extracellular side of the tunnel. The widened constriction regions indicate a pre-activated conformation exhibiting a closed channel with fractional transition to an open-channel state. Despite the similar tunnel profiles of the two protomers, the changes in chain A (*Figure 1B*) are more profound than in chain B (*Figure 1C*), particularly near C1, indicating conformational differences between the tunnels of the two chains.

### Bromide binding at the tunnel entry

A bromide ion was found at the cytoplasmic port of the tunnel in each protomer (*Figure 1A*). Based on the composite omit map, the signal of bromide in chain A is stronger than in chain B (*Figure 1—figure supplement 1A–B*). A priori, given the larger number of electrons in Br (Z = 35), it would be difficult to mistake it for a water molecule (Z = 10), but to test this possibility directly, the bromide ion was replaced with a water molecule and the structure refined using *PHENIX* (*Adams et al., 2010*). The refinement showed a strong positive electron density at the bromide position in the $F_o$-$F_c$ difference map (*Figure 1—figure supplement 1C–D*) and it was diminished only when a bromide ion was placed at that position (*Figure 1—figure supplement 1E–F*). This evidence excludes a water molecule as responsible for the electron density at the position. From these data and additional evidence presented below, we conclude a bromide ion resides at the tunnel entry in each protomer.

The bromide-binding site is formed by three cyclic residues: Pro58 from the cytoplasmic loop between TM1 and 2, and Trp246 and Trp250 from TM7 in a triangular configuration (*Figure 2A*). In the binding site, a bromide is stabilized by a H-bond interaction formed by the indole NH group of Trp250. This type of anion binding conformation has also been found in several chloride-bound nucleotide structures (*Auffinger et al., 2004*). Pro58 may play an important role in substrate binding by pressing its ring towards the bromide anion with a distance of 3.6 Å and 3.1 Å between the bromide ion and CG and CD atoms of Pro58 in chain A and 4.0 Å and 3.2 Å in chain B. Unlike other aromatic residues, the ring of proline exhibits a partial positive charge due to electron withdrawal by the adjacent protein backbone and the lower electronegativity of the hydrogens on the ring surface (*Zondlo, 2013*). The partial electropositivity of Pro58 may contribute to the binding of bromide via

**Table 1.** Crystallographic data and refinement of the bromide-bound GtACR1 structure*.

| PDB ID | 7L1E |
|---|---|
| Space group | $P\,2_1$ |
| a, b, c (Å) | 61.66, 77.64, 73.63 |
| α, β. γ (°) | 90, 95.59, 90 |
| Beamline | SLS-X06SA |
| Wavelength (Å) | 0.91882 |
| Resolution (Å) | 48.15–3.20 (3.28–3.20)[†] |
| Rmeas | 0.56 (2.81) |
| I /σ (I) | 2.84 (0.54) |
| Completeness (%) | 95.6 (96.0) |
| Multiplicity | 4.84 (3.31) |
| CC1/2 (%) | 97.5 (13.4) |
| Refinement | |
| Resolution (Å) | 41.73–3.2 (3.28–3.20) |
| No. of unique reflections | 11,054 (800) |
| Rwork/Rfree | 0.24/0.29 |
| R.m.s. deviations | |
| Bond lengths (Å) | 0.003 |
| Bond angles (°) | 0.730 |
| B-factor | |
| Proteins | 52.45 |
| Ligands | 51.31 |
| $H_2O$ | 33.96 |
| Ramachandran Plot | |
| Favored (%) | 96.18 |
| Allowed (%) | 3.82 |
| MolProbity Clash score | 12.22 |

*Data processing and refinement statistics are reported with Friedel pairs merged.

[†]Values in parentheses are for the highest resolution shell.

electrostatic interactions. Notably, this type of proline-halide interaction has also been observed in the structure of the chloride-pump rhodopsin ClR (site 2) (**Kim et al., 2016**).

Our previous results implicated Pro58 in gating of GtACR1, in that substitution of Pro58 by Glu reduced photocurrent amplitude, altered the kinetics of channel closing, and caused strong outward rectification of the current-voltage dependence (**Li et al., 2019**; **Sineshchekov et al., 2019**). We observed similar effects in W246E and W250E mutants in the bromide-binding site (**Figure 2B**), in that photocurrent amplitudes were significantly reduced compared to the wild type and outward rectification was increased (**Figure 2B**). None of these three substitutions with Glu reduced the selectivity for anions, as assessed by reversal potential measurements (**Figure 2B**). Based on the structure, the two tryptophan residues, Trp246 and Trp250, appear to have different roles in bromide binding; for example, compared to distant Trp246 moving its aromatic ring away from the bromide, Trp250 is close to the anion and forms an H-bond interaction between its indole NH and the bromide. In further examination of this bromide binding conformation, we found that W250F shows reduction of the current amplitude by 50%, whereas W246F behaves like WT (**Figure 2B**). These results are consistent with the structural observations in which Trp250, but not Trp246, stabilizes bromide via H-bond interaction.

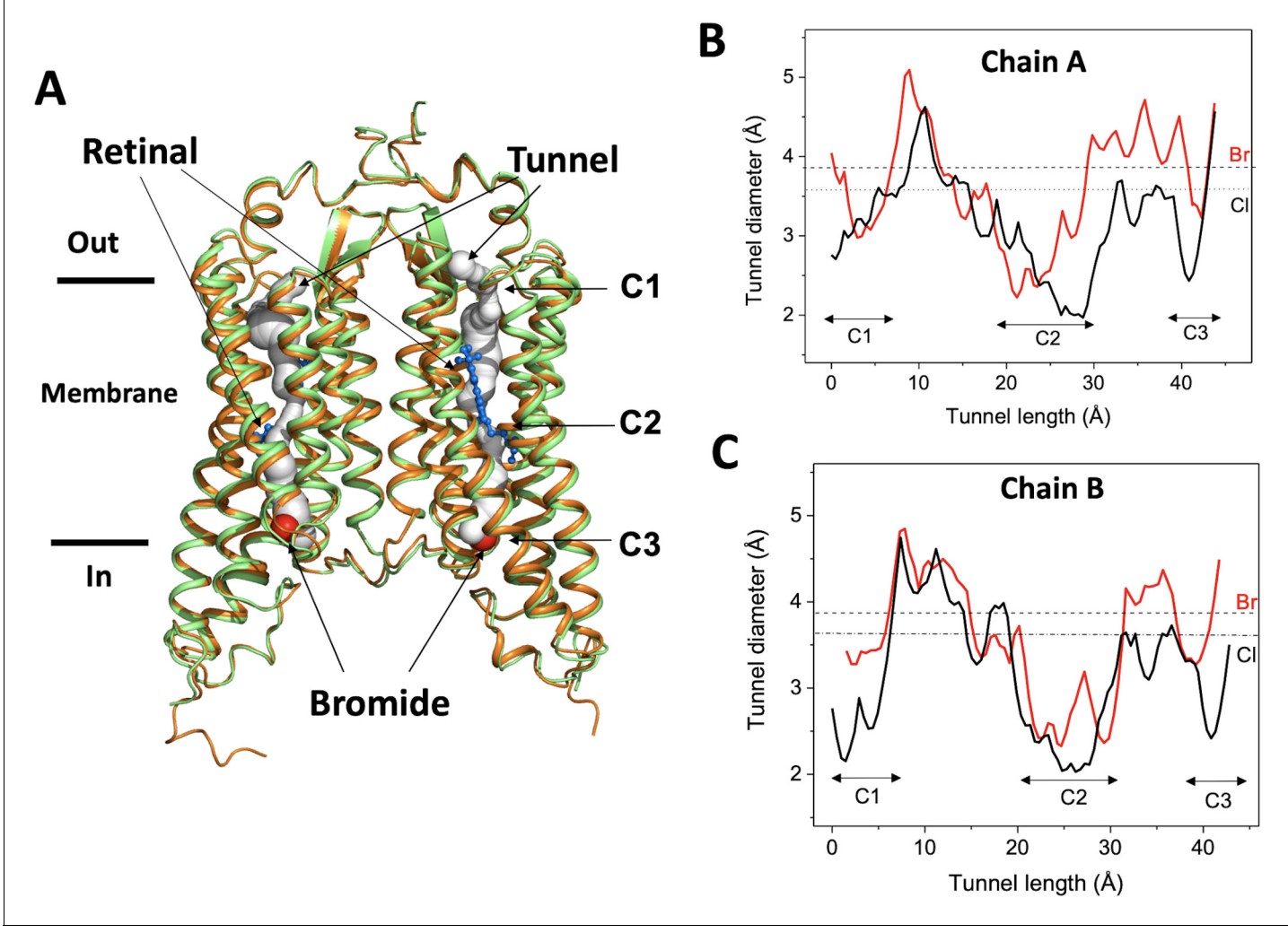

**Figure 1.** Overall conformation of the bromide-bound *Gt*ACR1 structure. (**A**) Superimposition of *Gt*ACR1 apo (*orange*, PDB 6EDQ) and bromide-bound (*green*) structures; one bromide ion (*red* sphere) is located at the cytoplasmic entry of the intramembrane tunnel (*grey* tube, predicted using the program CAVER [*Chovancova et al., 2012*] of each promoter). All-*trans*-retinal moieties are depicted as *blue* sticks. (**B–C**) Tunnel profiles of *Gt*ACR1 chain A (**B**) and chain B (**C**) predicted by CAVER: *Gt*ACR1 apo form (*black* line); bromide-bound form (*red* line). The sizes of chloride and bromide ions are indicated as dashed lines.

The online version of this article includes the following figure supplement(s) for figure 1:

**Figure supplement 1.** Confirmation of a bromide ion at the cytoplasmic port of *Gt*ACR1.

## Assesment of bromide binding by FTIR

To test for consequences of halide binding in the dark state with an alternative method, we investigated the effect of different halide anions on *Gt*ACR1 structure by measuring FTIR difference spectra at 80 K (see 'Materials and methods'). As previously shown, FTIR difference spectra of *Gt*ACR1 membranes recorded at 80 K reflect the photoinduced transition from the dark ground state to K and L-like intermediates formed upon illumination (*Yi et al., 2017*). Negative bands in the spectra measure vibrational frequencies of normal modes in the dark state structure, whereas positive bands measure photoinduced appearance of altered vibrational frequencies in the photoproducts. *Figure 3A* shows *Gt*ACR1 FTIR difference spectra recorded for *Gt*ACR1 multilamellar membrane films prepared using NaF, NaCl, or NaBr solutions. The major bands in the 1500–1600 cm$^{-1}$ region reflect the ethylenic (C=C) stretch mode of the retinylidene chromophore in *Gt*ACR1 (*Yi et al., 2017*). Changes in $\gamma_{C=C}$ in microbial rhodopsins entail alterations in the electron conjugation of the retinylidene polyene chain (*Bergo et al., 2004*; *Bergo et al., 2002*; *Bergo et al., 2003*; *Smith et al.,*

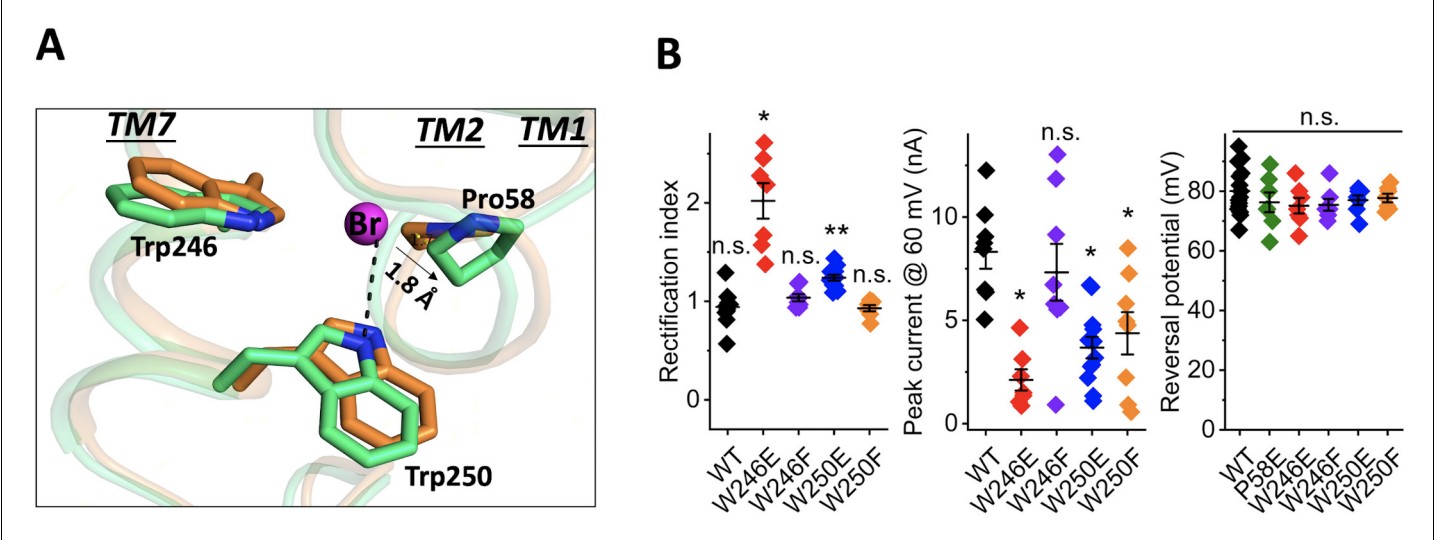

**Figure 2.** Structure of the bromide-binding site in the apo and bromide-bound *Gt*ACR1 and electrophysiological properties of site mutants. (A) A bromide ion (*magenta* sphere) stabilized by three cyclic residues (*green* sticks) via H-bond interaction (*black* dashed line) with superimposition of the apo form structure (*orange*). (B) Functional probing of the bromide-binding site residues by patch clamp analysis of their mutants: *Left*: Rectification index (RI), defined as the ratio of peak photocurrent amplitudes recorded at +60 and −60 mV at the amplifier output. RI > 1 by one-sample Wilcoxon signed-rank test: *p<0.05, **p<0.01, n.s. not significant (p>0.05). *Middle*: Peak current at 60 mV. Comparison with the wild-type by Mann-Whitney test: *p<0.005. *Right*: Reversal potential at the reduced Cl⁻ concentration in the bath. Comparison with the wild-type by Mann-Whitney test: n.s., not significant (p>0.05).

The online version of this article includes the following source data for figure 2:

**Source data 1.** Electrophysiological data for binding site mutants.

---

*1987a*). Notably, the frequency of the negative ethylenic band ($\gamma_{C=C}$) undergoes an upshift from 1529 cm$^{-1}$ in NaF to 1533 cm$^{-1}$ in NaBr (*Figure 3B*). The significant upward shift of 3 and 4 cm$^{-1}$ from that of Cl⁻ and F⁻, respectively, by Br⁻ in the dark state, provides compelling evidence for Br⁻ binding in the dark. This Br⁻ induced shift reveals an alteration in charge conjugation in the polyene chain of the retinylidene chromophore due to Br⁻ binding. In the case of NaBr the frequency upshift accentuates the appearance of a negative feature near 1520 cm$^{-1}$ that appears as a shoulder in the case of NaF and NaCl (*Figure 3B*). We conclude that Br⁻ interacts with *Gt*ACR1 in the unilluminated ground state.

While the different halides alter the retinylidene $\gamma_{C=C}$ band in the dark and L-like states, the all-*trans* to 13-*cis* isomerization that occurs upon light absorption and induces formation of the K and L-like intermediates appears to be unaffected. The unperturbed photoisomerization reaction, consistent with the photocurrent function observed with each of the three halides (*Govorunova et al., 2015*), is indicated by the absence of frequency shift differences in the negative and positive bands in the mixed C-C stretch fingerprint region from 1100 to 1300 cm$^{-1}$ for the different halides (*Figure 3*) since these band frequencies ($\gamma_{C-C}$) are highly sensitive to the retinal isomeric configuration (*Smith et al., 1987a*; *Smith et al., 1987b*).

## Conformational changes of the C1 and C3 constrictions

Despite the similar overall structure to that of the apo form, conformational changes were observed at the C1 and C3 constrictions within the tunnel. Pro58 is an important component of C3. In the apo form structure, Pro58, together with Leu108, Ala61, and Leu245, constrains the cytoplasmic port, leading to the cytoplasmic half of the tunnel narrowing to 3.6 Å in diameter (*Li et al., 2019*). In the bromide-bound structure, the presence of bromide pushes Pro58 outward by ~1.8 Å (*Figure 2A*). As a result, the cytoplasmic half of the tunnel is broadened by 1 Å in diameter in both protomers (*Figure 1B-C*).

Conformational changes were observed in the extracellular half of the tunnel between chain A and chain B (*Figure 1B–C*). In the apo form structure, the C1 constriction is stabilized by a salt-

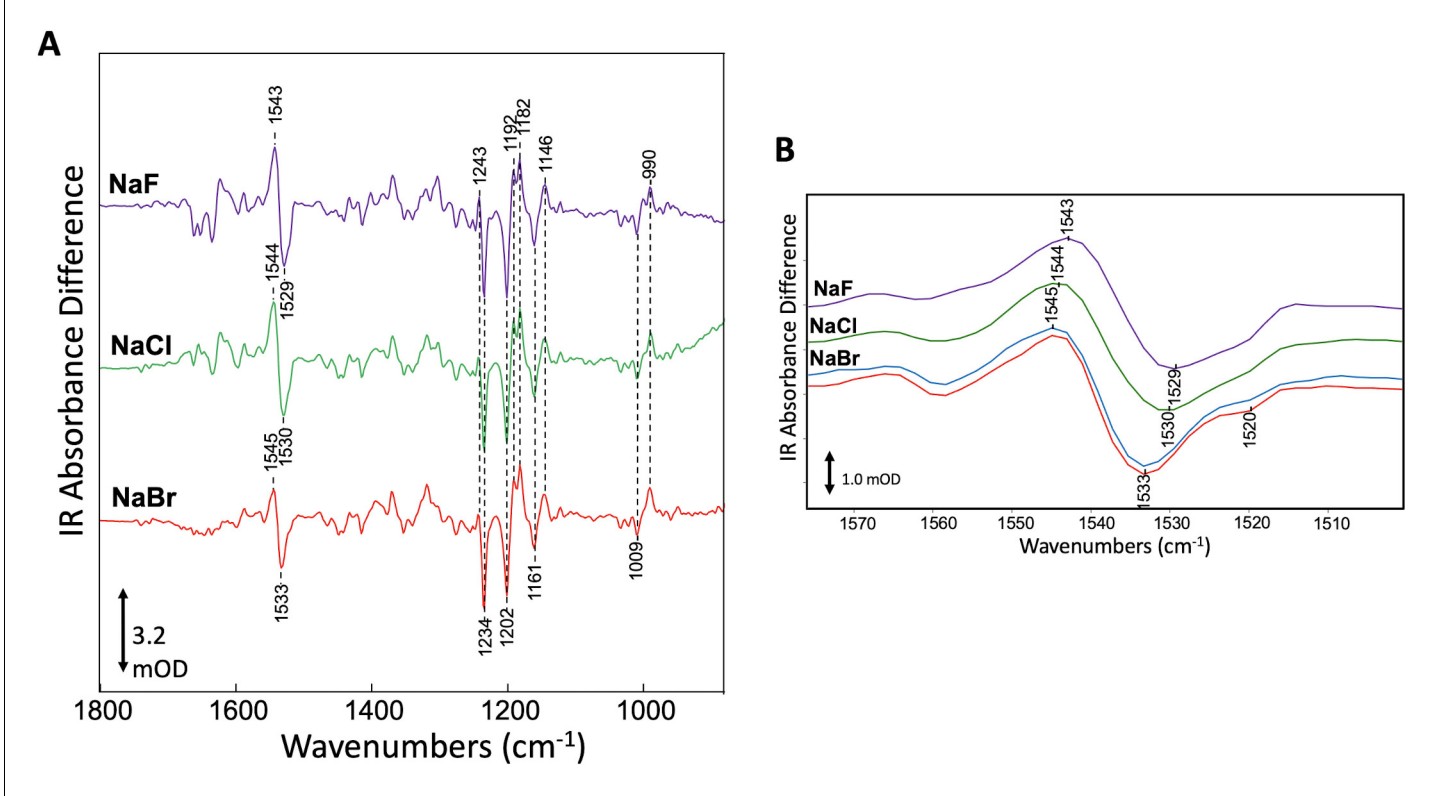

**Figure 3.** Specific interaction of bromide with *Gt*ACR1 detected by FTIR. (**A**) Comparison of FTIR difference spectra recorded at 80 K of *Gt*ACR1 hydrated multilamellar films formed using different halides. Spectra shown are the average of ≥5 individual difference spectra (see 'Materials and methods'). The scale bar shown is for the NaBr difference spectrum. (**B**) Comparison of FTIR difference spectra of *Gt*ACR1 in the ethylenic C=C stretch region. All spectra shown are from (**A**) (*purple*, *green*, and *red* plots) except the NaBr difference spectrum (*blue* plot) which was recorded on the same film after 3 days to evaluate the reproducibility of the FTIR measurement. The scale bar shown is for the NaBr difference spectrum (*red* plot).

bridge formed between Arg94 and Glu223 near the extracellular surface (*Li et al., 2019*). In the bromide-bound structure, chain B exhibits a similar conformation of C1 as that in the apo form. In contrast, Arg94 undergoes salt-bridge switching along the tunnel in chain A (*Figure 4A*). The side-chain of Arg94 is flipped by ~180° to form an alternative salt-bridge with Asp234 in the photoactive site (*Figure 4A*), resulting in modest relaxation (~1 Å in diameter) of the C1 constriction (*Figure 1B*).

Arg94 is highly conserved in the microbial rhodopsin family and it is critical for anion conductance of *Gt*ACR1. The mutation R94A or R94E nearly abolished anion conductance (*Li et al., 2019*; *Sineshchekov et al., 2019*). Arg94 is the only positively charged residue in the extracellular half of the tunnel. It may enable transfer of anions across the extracellular half of the tunnel via charge-charge interaction. We found that this side chain rotation enables Arg94 and its neighboring residues to form a conformation nearly identical to the chloride-binding site of the Cl⁻ pump ClR (*Figure 4B*). In the structure of ClR, a chloride ion (site 1) is bound between Arg95 and the Schiff base via H-bonds and salt-bridges. Based on the similar chemical conformations (*Figure 4B*), it is possible that Arg94 rotates its side chain to form an anion-binding site with the Schiff base in *Gt*ACR1.

Studies in three different laboratories have concluded that Asp234 is neutral in the dark state from measurements of the D234N mutant of *Gt*ACR1 by UV-vis absorption spectroscopy (*Kim et al., 2018*; *Sineshchekov et al., 2016*), resonance Raman spectroscopy (*Yi et al., 2016*), and FTIR (*Kim et al., 2018*). Both studies of independently determined crystal structures of *Gt*ACR1 attribute the major component of its neutralization to hydrogen-bonding to Tyr207 and Tyr72 (*Kim et al., 2018*; *Li et al., 2019*), leaving open partial electronegativity of Asp234 participating in H-bonding to the protonated Schiff base (PSB). The Asp234 residue is expected to be functionally important given its proximity to the PSB and its nearly universal conservation in microbial rhodopsins. *Kim et al.,*

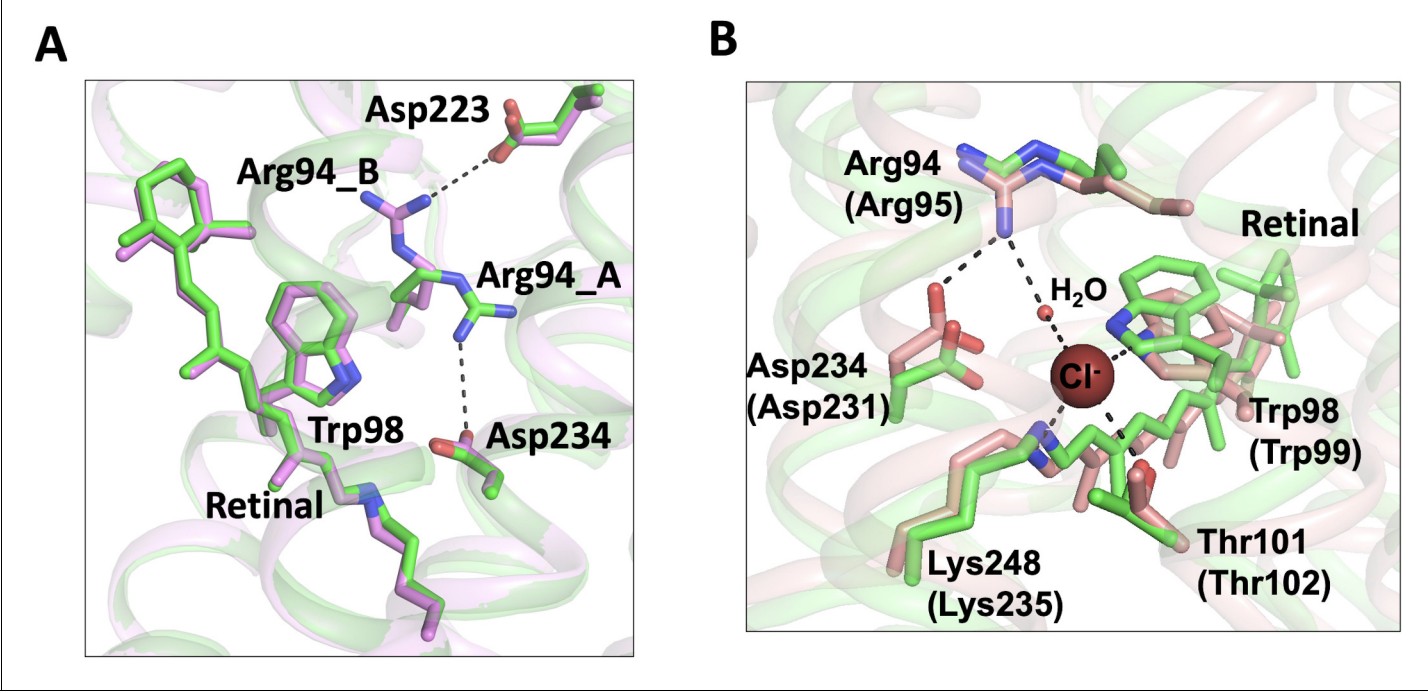

**Figure 4.** Conformational changes of the C1 constriction. (**A**) Superimposition of chain A (*green*) and chain B (*pink*) of the bromide-bound *Gt*ACR1 structure showing salt-bridge switching (*black* dashed lines) of Arg94 from the extracellular Asp223 to Asp234 near the mid-membrane retinal (*blue* sticks). Arg94_A and Arg94_B designate Arg94 in chains A and B, respectively. (**B**) Superimposition of chain A of the bromide-bound *Gt*ACR1 (*green*) and Cl-pump ClR (*red*, PDB: 5G2A) structures showing a putative halide-binding site in the extracellular half of the tunnel of *Gt*ACR1. In the ClR structure, a chloride ion (*maroon* sphere) is stabilized via H-bond interactions (*black* dashed lines). The residues are labelled as in *Gt*ACR1 and analogous residues for ClR are indicated in parentheses.

The online version of this article includes the following figure supplement(s) for figure 4:

**Figure supplement 1.** Comparison of Arg94 in *Gt*ACR1 and Arg95 in ClR.

*2018* conducted an extensive analysis of Asp234 and report that the D234N mutation nearly abolished photocurrents. Reduced photocurrents to 20% of wild-type from the D234N mutation were also observed by *Sineshchekov et al., 2015*. Differences in extent of photocurrent reduction are likely attributable to different assay conditions used in these studies. The electrostatic interaction of Arg94 with Asp234 in the pre-activated state may be correlated with the change in the electron conjugation of the retinylidene polyene chain in the dark that we observed by FTIR.

## Discussion

In this research advance, we addressed several major questions raised by our previous apo form structure by determining the crystal structure of bromide-bound *Gt*ACR1. We identified a novel bromide-binding site at the cytoplasmic entry of the transmembrane tunnel (*Figure 1*). The identification of bromide is supported by multiple lines of evidence: (1) the composite omit map indicates the presence of bromide at the cytoplasmic port (*Figure 1—figure supplement 1A–B*); (2) we exclude the possibility of a water at the bromide position as demonstrated in the $F_o$-$F_c$ difference map (*Figure 1—figure supplement 1C–D*); (3) the bromide-binding site exhibits a similar chemical conformation seen in chloride-binding structures (*Auffinger et al., 2004*); (4) functional analysis of W250F and W246F is consistent with the H-bond interaction in the bromide-binding site (*Figure 2B*); (5) specific interaction of *Gt*ACR1 with bromide in the unphotolyzed dark state was further demonstrated by FTIR analysis (*Figure 3*). In sum, these data confirm the bromide binding conformation in the structure.

These findings provide direct evidence for our hypothesis (*Li et al., 2019*) that the tunnel we observe both in the apo form and in an altered form in the bromide-bound condition, is the closed anion channel. The structure shows protein conformational changes in the bromide-bound tunnel that widen the tunnel: (1) bromide binding at the cytoplasmic entry induces relaxation of the C3 constriction (*Figure 2A*); and (2) salt-bridge switching of Arg94 widens the tunnel and may facilitate anion binding with the protonated Schiff base (*Figure 4A-B*).

The two protomers, chains A and B, share a very similar overall structure (rmsd = 0.5 Å). Both chains show a similar bromide binding conformation at the cytoplasmic entry and bromide binding to each site results in widening of the C3 constriction in each protomer (*Figure 1B–C*). The most significant difference is found in the tunnel at the Arg94 residue located at the C1 constriction (*Figure 4A*). In chain B, the conformation of Arg94 is nearly identical to that found in the apo form structure. Its side chain extends towards the extracellular side of the tunnel and is stabilized by a salt-bridge with Glu223. In contrast, Arg94 undergoes a salt-bridge switching in chain A; it rotates its side chain towards the retinylidene Schiff base to form an alternative salt-bridge with Asp234. This conformational change results in widening of the C1 constriction in chain A (*Figure 1B*). Based on these observations, we propose as a working hypothesis that the conformational differences of Arg94 between chains B and A accompanied by bromide binding at C3 demonstrate a transition from an inactivated state to a pre-activated state in the dark.

The conformational change of Arg94 near C1 is not likely to be directly induced allosterically by bromide binding at distant C3 since it is only observed in chain A, not in chain B. Instead, this conformational change may reflect the intrinsic flexibility property of Arg94 in the tunnel in the bromide-bound state. Weak binding and low occupancy of bromide at C1 is also a possibility suggested by the similar conformations of Arg94 of *Gt*ACR1 (chain A) and Arg95 of ClR (*Figure 4B*). These two counterpart residues appear to be stabilized by distinct H-bond networks. In *Gt*ACR1, inward Arg94 only forms a salt-bridge with Asp234 and an H-bond with a water molecule (*Figure 4—figure supplement 1A*). However, in the ClR structure, in addition to the salt-bridge, Arg95 is further stabilized by three polar residues, Asn92, Gln224, and Thr228, via two water molecules from the extracellular side of the protein (*Figure 4—figure supplement 1B*). The absence of these polar residues and waters in the vicinity may liberalize Arg94 and facilitate its flip-flopping in the tunnel of *Gt*ACR1.

Despite the conformational changes that expanded the tunnel in C1 and C3, C2 remains as the narrowest constriction attributable to the *trans*-configured retinal moiety within the tunnel (*Figure 1B*), suggesting a dominant role of the photoactive site Schiff base per se in the channel-gating mechanism. Moreover, the pre-activating conformational changes induced by bromide binding may facilitate channel opening by reducing free energy in the tunnel constrictions, consistent with the larger conductance of bromide vs chloride (*Govorunova et al., 2015*).

FTIR data show that bromide binding in the dark significantly alters the electron conjugation of the retinylidene polyene chain without perturbing the all-*trans* to 13-*cis* photoisomerization reaction occurring in the femtosec to picosec time window. Future structural study of *Gt*ACR1 in light-activated states is needed to resolve protein conformational changes in its photochemical reaction cycle.

## Materials and methods

**Key resources table**

| Reagent type (species) or resource | Designation | Source or reference | Identifiers | Additional information |
|---|---|---|---|---|
| Gene (*Guillardia theta*) | *Gt*ACR1 | Synthetic | GenBank: KP171708 | Humanized gene |
| Cell line (*Pichia pastoris*) | *Pichia pastoris* | Sigma Aldrich | Sigma Aldrich: 89070101, RRID:CVCL_0549 | Methylotrophic yeast |
| Cell line (*Homo sapiens*) | HEK293 | ATCC | ATCC: CRL-1573, RRID:CVCL_0045 | |

*Continued on next page*

*Continued*

| Reagent type (species) or resource | Designation | Source or reference | Identifiers | Additional information |
|---|---|---|---|---|
| Recombinant DNA reagent | Cellfectin II Reagent | Thermo Fisher | Cat. No.: 10362100 | https://www.thermofisher.com/order/catalog/product/10362100 |
| Recombinant DNA reagent | ScreenFectA transfection reagent | Waco Chemicals USA | Cat. No.: 29973203 | http://www.ereagent.com/uh/Shs.do?now=1544459665328 |
| Recombinant DNA reagent | pPIC9K | Thermo Fisher | Cat. No.: V175–20 | https://www.thermofisher.com/order/catalog/product/V17520#/V17520 |
| Recombinant DNA reagent | pcDNA3.1 | Thermo Fisher | Cat. No.: V79020 | https://www.thermofisher.com/order/catalog/product/V79020 |
| Software, algorithm | PyMol | PyMOL Molecular Graphics System, Schrödinger, LLC | RRID:SCR_000305 | http://www.pymol.org |
| Software, algorithm | UCSF Chimera | UCSF Resource for Biocomputing, Visualization, and Bioinformatics | RRID:SCR_004097 | http://plato.cgl.ucsf.edu/chimera |
| Software, algorithm | PHENIX | PMID:20124702 | RRID:SCR_014224 | http://www.phenixonline.org/ |
| Software, algorithm | Coot | PMID:15572765 | RRID:SCR_014222 | http://www.biop.ox.ac.uk/coot |
| Software, algorithm | OriginPro 2016 | OriginLab | | https://originlab.com |
| Software, algorithm | pClamp 10 | Molecular Devices | RRID:SCR_01123 | http://www.moleculardevices.com/products/software/pclamp.html |

## Protein expression and purification

Protein expression and purification of *Gt*ACR1 expressed in *Pichia pastoris* followed the procedure described (*Li et al., 2019*). The eluted protein was further purified using a Superdex Increase 10/300 GL column (GE Healthcare, Chicago, IL) equilibrated with buffer containing 350 mM NaBr, 5% glycerol, 0.03% DDM, 20 mM MES, pH 5.5, thereby replacing Cl$^-$ with Br$^-$ in the micelle suspension. Protein fractions with an A280/A515 absorbance ratio of ~1.9 were pooled, concentrated to ~20 mg/ml using a 100 K MWCO filter, flash-frozen in liquid nitrogen, and stored at −80°C until use. Molar protein concentration was calculated using the absorbance value at 515 nm divided by the extinction coefficient 45,000 M$^{-1}$cm$^{-1}$.

## Protein crystallization

Crystallization was carried out using the in meso method as with the apo protein (*Li et al., 2019*). The lipidic mesophase (lipidic cubic phase, LCP) sample was obtained by mixing 40 µl of *Gt*ACR1 protein with 60 µl monoolein (MO) (Sigma, St. Louis, MO; or Nu-chek, Waterville, MN) using two syringes until the mixture became transparent. Crystallization trials were set up using both 96-well LCP glass sandwich plates (Molecular Dimensions, Maumee, OH) and IMISX plates (MiTeGen), which are designed to perform in meso in situ serial X-ray crystallography (*Huang et al., 2018*; *Huang et al., 2016*) with 150 nl aliquots of the protein-mesophase mixture and overlaid with 1.5 µl of precipitant solution using a Gryphon crystallization robot (Art Robbins, Sunnyvale, CA). The plates were covered by aluminum foil to maintain a dark environment and incubated at room temperature. Bromide-bound *Gt*ACR1 was crystallized in essentially the same conditions and pH as previously used for the apo form (*Li et al., 2019*) except 100 mM NaBr bromide was supplied in both protein

purification and crystallization buffers. The most highly diffracting crystals of ~20 μm in size were obtained after one month from a crystallization screen containing 15% 2-methyl-2,4-pentanediol (MPD), 0.1 M NaBr, and buffer of 0.1 MES, pH 5.5, or Na-acetate, pH 5.5. LCP crystals from glass plate were harvested using micromesh loops (MiTeGen, Ithaca, NY), and the wells with crystals-laden LCP in IMISX plate were retrieved using a glass cutter and scissors and mounted using 3D-printed holders (*Huang et al., 2016*; *Huang et al., 2015*). All the samples were flash-cooled in liquid nitrogen without any additional cryoprotectant and stored in uni-pucks (MiTeGen, Ithaca, NY) for X-ray diffraction. An improvement over our previous work was setting up the crystallization in the IMISX plates and shipping the plates to PSI. This step prevents potential damage to the crystals during harvesting and facilitates high-throughput screening in the beamline.

## Data collection and processing

X-ray diffraction data collections were performed on protein crystallography beamlines X06SA-PXI at the Swiss Light Source (SLS), Villigen, Switzerland. Data were collected with a $10 \times 10$ μm² micro-focused X-ray beam of 13.49 keV (0.91882 Å in wavelength) at 100 K using SLS data acquisition software suites (DA+) (*Wojdyla et al., 2018*). Continuous grid-scans (*Wojdyla et al., 2018*) were used to locate crystals in frozen LCP samples both from conventional loop and IMISX samples (*Huang et al., 2016*). The crystals harvested on loop were collected by the rotation method with 0.2 s exposure time, 0.2° oscillation for data collection and 30° wedge for each crystals. The sample using IMISX setup was measured by an automated serial data collection protocol (CY+) as described (*Basu et al., 2019*) using the following parameters: 0.2° oscillation and 0.1 s exposure time for data collection with 10–20° wedge for each crystal. 217 data sets were collected using the EIGER 16M detector operated in continuous/shutterless data collection mode. Data were processed with XDS and scaled and merged with XSCALE (*Kabsch, 2010a*; *Kabsch, 2010b*). Isomorphic data sets were selected using XDSCC12 (*Assmann et al., 2020*). The selection is first performed using a multi-dimensional scaling procedure which identifies data sets with large non-isomorphism relative to clusters of other data sets, and then further selection based on the ΔCC1/2. The ΔCC1/2 represents the influence of a set of reflections on the overall CC1/2 of the merged data. By this mean, 31 IMISX data sets and five data sets obtained from loop-mounted samples were selected and then merged to a final complete data set yielding a 3.2 Å resolution structure. Data collection and processing statistics are provided in *Table 1*.

## Structure determination and analysis

The structure of bromide-bound *Gt*ACR1 was determined by the molecular replacement (MR) method using 6EDQ (*Li et al., 2019*) as the search model with the program Phaser (*McCoy et al., 2007*). The structure was refined using PHENIX (*Adams et al., 2010*) and model building was completed manually using COOT (*Emsley and Cowtan, 2004*). The final structure yields a $R_{work}/R_{free}$ factor of 0.26/0.29. Refinement statistics are reported in *Table 1*. The structure factors and coordinates have been deposited in the Protein Data Bank (PDB entry code: 7L1E). Figures of molecular structures were generated with PyMOL (http://www.pymol.org). We analyzed the halide tunnel using the program CAVER with 0.9 Å as the detecting probe (*Chovancova et al., 2012*).

## Electrophysiology of *Gt*ACR1 mutants

*Gt*ACR1 mutants were characterized by whole-cell patch clamp recording as described in detail in our previous report (*Li et al., 2019*). Briefly, the wild-type expression construct was cloned into the mammalian expression vector pcDNA3.1 (Life Technologies, Carlsbad, CA) in frame with an EYFP (enhanced yellow fluorescent protein). Mutations were introduced using a QuikChange XL site-directed mutagenesis kit (Agilent Technologies, Santa Clara, CA) and verified by DNA sequencing. HEK293 (human embryonic kidney) cells were transfected using the ScreenFectA transfection reagent (Waco Chemicals USA, Richmond, VA). All-*trans*-retinal (Sigma, St. Louis, MO) was added at the final concentration 4 μM immediately after transfection. Photocurrents were recorded 48–72 hr after transfection in whole-cell voltage clamp mode at room temperature (25°C) with an Axopatch 200B amplifier (Molecular Devices, Union City, CA) and digitized with a Digidata 1440A using pClamp 10 software (both from Molecular Devices). Patch pipettes were fabricated from borosilicate glass and filled with the following solution (in mM): KCl 126, MgCl$_2$ 2, CaCl$_2$ 0.5, EGTA 5, HEPES 25,

and pH 7.4. The standard bath solution contained (in mM): NaCl 150, CaCl$_2$ 1.8, MgCl$_2$ 1, glucose 5, HEPES 10, pH 7.4. To test for changes in the permeability for Cl$^-$, this ion in the bath was partially replaced with non-permeable aspartate (the final Cl$^-$ concentration 5.6 mM). For each cell, a single value of the E$_{rev}$ was calculated. The holding potential values were corrected for liquid junction potentials calculated using the Clampex built-in LJP calculator. Continuous light pulses were provided by a Polychrome V light source (T.I.L.L. Photonics GMBH, Grafelfing, Germany) at 15 nm half-bandwidth in combination with a mechanical shutter (Uniblitz Model LS6, Vincent Associates, Rochester, NY; half-opening time 0.5 ms). The maximal light intensity at the focal plane of the objective lense was 7.7 mW mm$^{-2}$ at 515 nm.

Batches of culture were randomly allocated for transfection with a specific mutant; no masking (blinding) was used. Individual transfected HEK293 cells were selected for patching by inspecting their tag fluorescence; non-fluorescent cells were excluded, as were cells for which we could not establish a gigaohm seal. Results obtained from different individual cells were considered as biological replicates. The raw data obtained in individual cells are shown as diamonds. Sample size was estimated from previous experience and published work on a similar subject, as recommended by the NIH guidelines. No outliers were excluded. Normality of the data was not assumed, and therefore non-parametric statistical tests were used as implemented in OriginPro 2016 software; P values > 0.05 were considered not significant.

## Cell lines
Only a commercially available cell line authenticated by the vendor (HEK293 from ATCC) was used; no cell lines from the list of commonly misidentified cell lines were used. The absence of micoplasma contamination was verified by Visual-PCR mycoplasma detection kit (GM Biosciences, Frederick, MD).

## FTIR difference spectroscopy
The expression, purification, and reconstitution of GtACR1 into proteolipid membranes using E. coli polar lipids was performed as described previously (*Yi et al., 2017*). The GtACR1 membranes were washed and centrifuged 5x in 1M NaF, NaCl and NaBr solutions and the pellet finally resuspended in a solution of 100 mM NaF, NaCl, or NaBr. Multilamellar films were formed by drying a small drop of this suspension on a BaF window which is rehydrated with a drop of water in a sealed cell. FTIR difference measurements at 80 K were performed as previously described (*Yi et al., 2017*). All spectra were recorded using a Bio-Rad FTS-60A FTIR spectrometer (Agilent Technologies, Inc) equipped with a liquid nitrogen cooled HgCdTe detector. Each acquired spectrum consisted of 200 scans (approximately one-minute total acquisition time) recorded at 4 cm$^{-1}$ resolutions. Difference spectra consisted of an average of at least five difference spectra recorded immediately before illumination each subtracted from a spectrum recorded during illumination using 505 nm LED light. In order to repeat this measurement and average multiple differences, the sample was photoreversed after illumination with 590 nm light (*Yi et al., 2017*).

## Acknowledgements
This work was supported by National Institutes of Health Grants R01GM027750 and R35GM140838 and Endowed Chair AU-0009 from the Robert A Welch Foundation to JLS, American Heart Association Grant 18TPA34230046 to LZ, and the National Science Foundation Division of Chemical, Bioengineering, Environmental and Transport Systems Grant CBET-1264434 to KJR. C-YH was partially supported by the European Union's Horizon 2020 research and innovation programme under the Marie-Skłodowska-Curie grant agreement No. 701647. The authors thank Yumei Wang for her technical assistance and the assistance and support of beamline scientists at the Swiss Light Source beamlines X06SA-PXI.

## Additional information

### Competing interests

Elena G Govorunova, Oleg A Sineshchekov, John L Spudich: as an inventor and The University of Texas Health Science Center at Houston has been granted a patent titled: Compositions and Methods for Use of Anion Channel Rhodopsins Patent # 10,519,205 granted Dec 31, 2019 by the US Patent and Trademark Office. The other authors declare that no competing interests exist.

### Funding

| Funder | Grant reference number | Author |
| --- | --- | --- |
| National Institute of General Medical Sciences | R01GM027750 | John L Spudich |
| National Institute of General Medical Sciences | R35GM140838 | John L Spudich |
| Welch Foundation | Endowed Chair AU-0009 | John L Spudich |
| American Heart Association | 18TPA34230046 | Lei Zheng |
| National Science Foundation | CBET-1264434 | Kenneth J Rothschild |
| Horizon 2020 | 701647 | Chia-Ying Huang |

The funders had no role in study design, data collection and interpretation, or the decision to submit the work for publication.

### Author contributions

Hai Li, Chia-Ying Huang, Elena G Govorunova, Data curation, Formal analysis, Investigation, Methodology, Writing - original draft, Writing - review and editing; Oleg A Sineshchekov, Data curation, Formal analysis, Investigation, Methodology, Writing - original draft; Adrian Yi, Kenneth J Rothschild, Data curation; Meitian Wang, Lei Zheng, Conceptualization, Data curation, Formal analysis, Supervision, Investigation, Methodology, Writing - original draft, Writing - review and editing; John L Spudich, Conceptualization, Data curation, Formal analysis, Supervision, Investigation, Methodology, Writing - original draft, Project administration, Writing - review and editing

### Author ORCIDs

Hai Li http://orcid.org/0000-0002-3969-6709
Chia-Ying Huang http://orcid.org/0000-0002-7676-0239
Elena G Govorunova http://orcid.org/0000-0003-0522-9683
Lei Zheng https://orcid.org/0000-0001-7789-5234
John L Spudich https://orcid.org/0000-0003-4167-8590

### Decision letter and Author response

Decision letter https://doi.org/10.7554/eLife.65903.sa1
Author response https://doi.org/10.7554/eLife.65903.sa2

## Additional files

### Supplementary files

• Transparent reporting form

### Data availability

Diffraction data have been deposited in PDB under the accession code 7L1E.

The following dataset was generated:

| | Database and |
| --- | --- |

| Author(s) | Year | Dataset title | Dataset URL | Identifier |
|---|---|---|---|---|
| Li H, Huang C-Y, Spudich JL, Zheng L | 2021 | The Crystal Structure of Bromide-Bound GtACR1 | https://www.rcsb.org/structure/7L1E | RCSB Protein Data Bank, 7L1E |

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
