## [Decision Letter]

**Acceptance summary:**

This manuscript reports a significant contribution towards an improved mechanistic understanding of light gated anion channels. The authors' findings show that bromide binding induces a transition from an inactivated state to a pre-activated state in the dark that facilitates channel opening. The studies provide a basis for optimizing the light-gated anion channel GtACR1 as an optogenetic tool.

**Decision letter after peer review:**

Thank you for submitting your article "The Crystal Structure of Bromide-bound *Gt*ACR1 Reveals a Pre-Activated State in the Transmembrane Anion Tunnel" for consideration by *eLife*. Your article has been reviewed by 2 peer reviewers, and the evaluation has been overseen by Sriram Subramaniam as Reviewing Editor and Richard Aldrich as the Senior Editor. The reviewers have opted to remain anonymous.

Essential revisions:

1. Several differences are observed between chains A and B in the coordinates. Wherever possible, the authors should present the structures of both chain A and B in the main and supplementary figures.

2. Line237-239: Please explain the logic for merging the datasets collected from crystals in the IMISX setup and from crystals harvested on the loop.

3. Please use the dataset collected at 0.91882Å to draw an anomalous difference Fourier map. If the map does not support the bromide binding, then other experiments such as ITC or FTIR would be needed to validate bromide binding in the dark.

4. Please add a brief discussion on the change in C1 being an allosteric effect of bromide binding near C3 vs. the result of a low bromide occupancy at C1.

5. Include relevant citations to previously published work that also identified Asp 234 as a critical amino acid that is protonated in the dark state.

---

## [Author Response]

Essential revisions:1. Several differences are observed between chains A and B in the coordinates. Wherever possible, the authors should present the structures of both chain A and B in the main and supplementary figures.

In the revision we present the structures side-by-side of both chains A and B in the main and supplementary files. We thank the editors for pointing out that we should show more clearly the differences between the two protomers, which raised interesting aspects of the structure worthy of discussion. The two chains share a very similar overall structure (RMSD = 0.5Å). Both chains A and B show a similar bromide binding conformation at the cytoplasmic entry and bromide binding to each site results in widening of the C3 constriction in each protomer (Figure 1B and 1C). The identification of bromide in each chain is demonstrated in Suppl. Figure 1. The most significant difference is found in the tunnel at the Arg94 residue located at the C1 constriction (Figure 4A). In chain B, the conformation of Arg94 is nearly identical to that found in the apo form structure. Its side chain extends towards the extracellular side of the tunnel and is stabilized by a salt-bridge with Glu233. In contrast, Arg94 undergoes a salt-bridge switching in chain A; it rotates its side chain towards the retinylidene Schiff base to form an alternative salt-bridge with Asp234. This conformational change results in widening of the C1 constriction in chain A (Figure 1B). We have added discussion of these differences and their implications in the Discussion section (lines 255-266).

2. Line237-239: Please explain the logic for merging the datasets collected from crystals in the IMISX setup and from crystals harvested on the loop.

We explain the logic here and in the revision the details are in the Materials and Methods section (lines 327-345). Data collection was carried out using both loop-mounted crystals and the IMISX setup. We initially performed data collection using crystals mounted on the loop. To avoid radiation damage, only a small wedge dataset was collected from each Br-*Gt*ACR1 crystal. To overcome the limited amount of data from each wedge, we then applied the IMISX method in order to screen a large number of crystals efficiently. We were able to collect 217 small wedges data sets using the IMISX setup. These datasets were carefully selected using XDSCC12 *(Assmann et al., 2020*) to eliminate non-isomorphism. The selection was first performed as a multi-dimensional scaling procedure that identifies data sets with large non-isomorphism relative to clusters of other datasets, and then further selects based on the ΔCC_1/2_. The ΔCC_1/2_ represents the influence of a set of reflections on the overall CC_1/2_ of the merged data. By this means, 31 isomorphic small wedge data sets were selected. To improve data completeness, we also selected 5 isomorphic datasets from loop samples. The total 36 small wedge datasets were merged using XDS to yield a final 3.2 Å complete data set.

3. Please use the dataset collected at 0.91882Å to draw an anomalous difference Fourier map. If the map does not support the bromide binding, then other experiments such as ITC or FTIR would be needed to validate bromide binding in the dark.

The datasets were collected at 0.91882 Å wavelength, but we did not detect any strong bromide signals in the anomalous difference Fourier map. This may be due to preferential orientation of the thin-plate *Gt*ACR1 crystals in the IMISX plate. The weak Br signals may also be attributed to the weak bromide binding conformation, its partial occupancy, and poor intrinsic order. It is not unusual that anomalous signals are influenced by the location of the scatter. For example, in our previous structural determination of YfkE (Wu et al., PNAS 2013), Seleno-methionine was used to label 12 native Met residues. However, we could identify only 10 Se positions and the other 2 Se were undetectable in the anomalous difference map, despite the dataset collection at the Se absorption peak wavelength. Therefore, the lack of strong anomalous signals is not sufficient to exclude the presence of bromide in the structure.

During refinement, we carefully compared the new structure with our previous apo form since both structures were obtained in essentially the same conditions except for the replacement of NaCl with NaBr. We found an unusual density at an entry port to the tunnel. The structure could be better refined only if we included bromide at that position. Further supporting our conclusion, we present several lines of evidence confirming the presence of bromide: (1) We show the composite omit map calculated by PHENIX. As seen in Suppl. Figure 1A-1B, the omit map of bromide is contoured at 2σ in chain A and at 1.5σ in chain B. (2) We exclude the possibility of a water at the bromide position. PHENIX refinement with a water substitution showed a strong positive density in the *F_o_*-*F_c_* difference map (Suppl. Figure 1C-1D), which is diminished when refining a bromide at the position (Suppl. Figure 1E-1F). (3) A similar halide binding geometry is found in other halide-binding structures in microbial rhodopsins. (4) We have added FTIR data (Figure 3) showing specific interaction of bromide with *Gt*ACR1 in the dark, with effects different from other halides. (5) We included additional functional analysis of W250F and W246F confirming the specific H-bond interaction between bromide and W250, but not with W246, as predicted by the binding site (Figure 2B). Considering all of this evidence, we are confident in the assignment of the Br^-^ positions in the structure. We have addressed these data further in the Discussion section (lines 239-248).

Following the Editor’s suggestion, we applied a different method, FTIR, and validated bromide binding to *Gt*ACR1 in the dark state. We analyzed FTIR spectra of *Gt*ACR1 in the presence of 3 different halide ions: F^-^, Cl^-^, and Br^-^. Differences in major bands in the 1500-1600 cm^-1^ region that reflect the ethylenic (C=C) stretch mode of the retinylidene chromophore show a large bromide-induced alteration in the electron conjugation of the retinylidene polyene chain in the dark, confirming that bromide causes a significant structural change. We have added the FTIR data in the revision (Figure 3) and analysis in the main text, and have added as authors Prof. Kenneth Rothschild, a well-known expert molecular spectroscopist in the rhodopsin field, and his coworker Dr. Adrian Yi.

4. Please add a brief discussion on the change in C1 being an allosteric effect of bromide binding near C3 vs. the result of a low bromide occupancy at C1.

We thank the editors for raising this issue. In the revision we address it in Discussion (lines 267-278) put in after the discussion of the differences between chain A and chain B conformations as follows:

“The conformational change of Arg94 near C1 is not likely to be directly induced allosterically by bromide binding at distant C3 since it is only observed in chain A, not in chain B. […] The absence of these polar residues and waters in the vicinity may liberalize Arg94 and facilitate its flip-flopping in the tunnel of *Gt*ACR1.”

5. Include relevant citations to previously published work that also identified Asp 234 as a critical amino acid that is protonated in the dark state.

We have added the following paragraph at the end of the Results section (lines 220-234):

“Studies in 3 different laboratories have concluded that Asp234 is neutral in the dark state from measurements of the D234N mutant of *Gt*ACR1 by UV-vis absorption spectroscopy (Kim et al., 2018; Sineshchekov et al., 2016), Resonance Raman spectroscopy (Yi et al., 2016), and FTIR (Kim et al., 2018). […] The electrostatic interaction of Arg94 with Asp234 in the pre-activated state may be correlated with the change in the electron conjugation of the retinylidene polyene chain in the dark that we observed by FTIR.”